# Effects of Vitamin D Supplementation on Haematological Values and Muscle Recovery in Elite Male Traditional Rowers

**DOI:** 10.3390/nu10121968

**Published:** 2018-12-12

**Authors:** Juan Mielgo-Ayuso, Julio Calleja-González, Aritz Urdampilleta, Patxi León-Guereño, Alfredo Córdova, Alberto Caballero-García, Diego Fernandez-Lázaro

**Affiliations:** 1Department of Biochemistry, Molecular Biology and Physiology, Faculty of Physical Therapy, University of Valladolid, 42004 Soria, Spain; a.cordova@bio.uva.es; 2Department of Physical Education and Sports, University of Basque Country (UPV-EHU), 01007 Vitoria, Spain; julio.calleja.gonzalez@gmail.com; 3Elikaesport, Nutrition, Innovation & Sport, 08290 Barcelona, Spain; a.urdampilleta@drurdampilleta.com; 4Faculty of Psychology and Education, University of Deusto, Campus of Donostia-San Sebastián, 20012 San Sebastián, Guipúzcoa, Spain; patxi.leon@deusto.es; 5Department of Anatomy and Radiology, Faculty of Physical Therapy, University of Valladolid, 42004 Soria, Spain; albcab@ah.uva.es; 6Department of Cell Biology, Histology and Pharmacology, Faculty of Physical Therapy, University of Valladolid, 42004 Soria, Spain, diego.fernandez.lazaro@uva.es

**Keywords:** strength-endurance, vitamin D, hemoglobin, hematocrit, recovery, testosterone, cortisol

## Abstract

Introduction: Deficient levels of 25-hydroxyvitamin D (25(OH)D) (<30 ng/mL) may compromise health and athletic performance. Supplementation with oral vitamin D can favor the state of iron metabolism, and testosterone and cortisol as an indicator of muscle recovery of the athlete with a deficiency. The main aim of this study was to evaluate the influence of eight weeks of supplementation with 3000 IU/day of vitamin D on the hematological and iron metabolism profile, as well as on the analytical values of testosterone and cortisol on elite male traditional rowers. The secondary aim was to examine if serum 25(OH)D is a predictor of testosterone and cortisol levels. Material and Methods: Thirty-six elite male rowers (27 ± 6 years) were assigned to one of the two groups randomly: 1) Control group (CG, *n* = 18, height: 181.05 ± 3.39 cm and body mass: 77.02 ± 7.55 kg), 2) Group treated with 3,000 IU of vitamin D3/day (VD3G, *s* = 18, height: 179.70 ± 9.07 cm and body mass: 76.19 ± 10.07 kg). The rowers were subjected to blood tests at the beginning of the study (T1) and after eight weeks of treatment (T2), for the analysis of hematological and hormonal values. Repeated-measures ANOVA with group factor (GC and GVD3) were used to examine if the interaction of the different values was the same or different between the groups throughout the study (time × group) after vitamin D3 treatment. To analyze if 25(OH)D was a good predictor of testosterone, cortisol, and testosterone/cortisol ratio a stepwise regression model was performed. Results: Statistically significant and different increases were observed in the group-by-time interaction of 25(OH)D in VD3G in respect to CG during the study (*p* < 0.001; VD3G (T1: 26.24 ± 8.18 ng/mL vs. T2: 48.12 ± 10.88 ng/mL) vs CG (T1: 30.76 ± 6.95 ng/mL vs. T2: 35.14 ± 7.96 ng/mL). Likewise, significant differences between groups were observed throughout the study in the group-by-time interaction and changes of hemoglobin (GC: −2.89 ± 2.29% vs. VD3G: 0.71 ± 1.91%; *p* = 0.009), hematocrit (CG: −1.57 ± 2.49% vs. VD3G: 1.16 ± 1.81%; *p* = 0.019) and transferrin (CG: 0.67 ± 4.88% vs. VD3G: 6.51 ± 4.36%; *p* = 0.007). However, no differences between groups were observed in the group-by-time interaction of the hormonal parameters (*p* > 0.05). Regression multivariate analysis showed that cortisol and testosterone levels were associated with 25(OH)D levels (*p* < 0.05). Conclusion: Oral supplementation with 3000 IU/day of vitamin D3 during eight weeks showed to be sufficient to prevent a decline in hematological levels of hemoglobin and hematocrit, and improve transferrin of 25(OH)D levels. However, although it was not sufficient to enhance muscle recovery observed by testosterone and cortisol responses, it was observed that serum 25(OH)D levels could be a predictor of anabolic and catabolic hormones.

## 1. Introduction

Vitamin D is receiving wide attention in athletic communities given that non-optimal 25-hydroxyvitamin D (25(OH)D) values are a prevalent issue among athletes [1,2]. In this sense, it has been observed that deficient values of 25(OH)D may reduce physical performance among others due to inadequate recovery [3]. However, optimal levels of serum 25(OH)D are positively related to strength and potency [4], running performance [5], endurance performance [6], and aerobic capacity [7]. In this way, it has been observed that maximum peak performance coincides with the time of year that athletes have higher serum levels of 25(OH)D [7].

Although vitamin D is found naturally in only a few foods, such as fatty fish (i.e., mackerel, salmon, sardines), egg yolks, certain mushrooms, dairy products, margarine, ready-to-eat cereals, and fruit juices that have been fortified [8], sun exposure is the main source of vitamin D. Due to the low synthesis of vitamin D during some periods of the year, oral vitamin D supplementation has been proposed as an alternative to obtain adequate levels of 25(OH)D [9]. In this regard, a dose of 3000 IU/day of vitamin D over a period of eight weeks has been shown to adjust the concentration of 25(OH)D to 30–100 ng/mL (optimal range) in a general population with low levels [10,11]. However, optimal sport benefits occur at 25(OH)D levels above the current definition of sufficiency (>30 ng/mL) with no reported sports health benefits above 50 ng/mL [3].

A mechanism by which optimal 25(OH)D levels could enhance athletic performance is improving hematological levels. In this line, vitamin D has been proposed as a suitable factor that facilitates erythropoiesis, given that erythrocyte precursor cells express receptors of the active form of vitamin D, which induce the proliferation and maturation of erythrocytes [12]. Therefore, vitamin D deficiency could accelerate the decrease in hemoglobin and increase the incidence of anemia [13]. Thus, an inverse association of 25(OH)D levels with the risk of anemia has been shown [14]. However, there are conflicting data regarding the possible improvement of hematological levels with vitamin D supplementation in different populations [15]. Specifically, in the sports field, although there is little research, they conclude the existence of a positive relationship between 25(OH)D levels and iron status [16,17].

Another mechanism that promotes sport performance is adequate muscle recovery of athletes [18]. Although the parameters used to know their state are different, analysis of anabolic–catabolic hormones has shown a great relationship with the endogenous regenerative process of athletes [19,20]. Therefore, a low testosterone level and/or a high cortisol value could indicate inadequate muscle recovery [21]. In this sense, in the sports sciences field, Lombardi et al. [22] observed a positive and close correlation between 25(OH)D levels and testosterone, and inversely with those of cortisol. This detail could indicate that increasing the circulating levels of 25(OH)D in athletes through a supplementation with vitamin D could improve the levels and ratios of these hormones, thus facilitating recovery process. In this regard, supplementation with 3,332 IU/day showed significant increases in testosterone levels in non-athletes [23]. However, to our knowledge, no such data have been found in the athlete population.

For this reason, the main goal was to evaluate the influence of eight weeks of supplementation with 3,000 IU/day of vitamin D3 on the hematological and iron metabolism profile, as well as in the analytical values of testosterone and cortisol in elite traditional rowers. In addition, a secondary aim was to examine if serum 25(OH)D is a predictor of testosterone and cortisol levels.

## 2. Materials and Methods

### 2.1. Participants

Thirty-six elite male rowers (27 ± 6 years) who belonged to a rowing club from the First Trainer League in Spain (ACT) participated in this double-blind, placebo-controlled study. The tests were carried out in the spring months (from 8 April to 3 June 2018) with an average solar irradiance of 213.1 W/m^2^. The solar irradiance and outdoor air temperature were recorded every 10 min during the whole study by wireless Vantage Pro2™ Plus (Davis Instruments, Hayward, CA, USA). All participants were located in the area of San Sebastian (Guipúzcoa, Spain) at a latitude of 43°18′46″ N and with 20 km between the most distant cities. All rowers followed the same training program conducted and supervised by the same certificated strength and conditioning coach. The average weekly hours of training were 15 during study (50% strength training, 45% endurance training, and 5% complementary training). Further, the dietitian-nutritionist of the club elaborated an individual diet for each rower. The diets were proposed using previously established energy and macronutrient guidelines for adequate athletic performance [24], and were based on the volume and training load, and personal characteristics of each participant. Importantly, diets were described so that all athletes would meet the micronutrient recommended dietary allowances (RDAs) for men aged 19–30 [25,26].

A medical examination was performed before the study began in order to verify that the participants did not have any disease. In addition, no rower was taking medications that affected body mass, or any supplement or medication that affected analytical or performance values. The experimental procedures, associated risks, and benefits were explained prior to the medical examination, so each rower signed a written consent form prior to commencing study participation. None of the rowers had pre-existing injuries before or during the intervention period. This study was designed in accordance with the Declaration of Helsinki (2008) and Fortaleza update (2013) and approved by the ethics committee of the University of Basque Country (M10_2017_247).

### 2.2. Experimental Protocol and Evaluation Program

This study was structured with a double-blind, placebo-controlled design in order to analyze the effects of and 8-week oral supplementation of 3000 IU per day of vitamin D3 (Lindens Health + Nutrition Ltd., Wakefield, WF2) on the hematological, iron metabolism, and muscle recovery studied by catabolic–anabolic hormones. Participants were assigned to groups using a stratified block design. An independent statistician generated the random allocation sequence: 1) Control group (CG, *n* = 18, height: 181.05 ± 3.39 cm and body mass: 77.02 ± 7.55 kg), 2) Group treated with 3000 IU/day of vitamin D3 (VD3G, *n* = 18, height: 179.70 ± 9.07 cm and body mass: 76.19 ± 10.07 kg).

All participants attended the laboratory (08:30) for blood collection at two specific points during the study: 1) at baseline (T1), and 2) post-treatment (T2—the day after 8 weeks of treatment). The VD3G took 3000 IU of vitamin D3 with one capsule per day. While in the GC, the rowers assigned took capsules of similar external appearance filled with 10 mg of maltodextrin. Both groups took their dose every morning with their breakfast from the day following T1 to T2 (during 8 weeks). The control group served as baseline or “standard” condition because this group did not take any vitamin D3.

### 2.3. Blood Collection

Antecubital venous blood samples were collected from all participants to evaluate hematological, iron metabolism, and hormonal parameters in T1 and T2. All samples were examined under basal conditions after a night and after at least 36 h without exercise. At both times the rowers arrived at the laboratory at 08:30 where they were allowed to rest in a seat for 30 min, at which time the blood samples were recollected.

In T1 and T2, 22 mL of blood were taken. The first 13 mL were collected in a tube containing 200 μL EDTA K anticoagulant (Vacutainer, Becton Dickinson) and used to determine serum iron, ferritin, and transferrin. Serum concentrations of 25(OH)D were determined by HPLC-MS/MS using an AB Sciex 5500 tandem mass spectrometer (AB Sciex UK Ltd., Warrington, UK). The second tube with the remaining 9 mL collected separately was centrifuged for 15 min at 4 °C and 3000 rpm. After centrifugation, the serum was separated and stored in aliquots at −20 °C until analysis. A STKS (Coulter) analyzer was used to determine the content of hematological parameters.

Parameters relative to iron metabolism were measured using a COBAS FARA analyzer (Roche Diagnostics, Basel, Switzerland). Serum iron was determined using the ferrozine colorimetric method but without protein precipitation, while ferritin was measured by immunoturbidimetry.

As for the hormonal variables, commercially available enzyme immunoabsorbent assay kits (DRG testosterone ELISA kit^®^, DRG Instruments GmbH, Marburg/Lahn, Germany) were used for the measurement of serum total testosterone. The intra-assay coefficient of variation (CV) was 4.3% and the CV among the trials was 9.2%. For the measurement of serum cortisol levels, an enzyme-linked fluorescent assay with the aid of a multiparametric analyzer (Minividas^®^, Biomerieux, Marcy l’Etoile, France) was used. The substrate, 4-methyl umbelipherone, was used and performed a fluorescence emission at 450 nm, after stimulation at 370 nm. The intra-assay CV was 5.7% and the CV of the intermediate assay was 6.2%. Finally, the testosterone/cortisol ratio (TT/C) was calculated from the concentrations of testosterone and cortisol, dividing testosterone between cortisol.

### 2.4. Dietary Assessment

All athletes were informed on proper food tracking by trained and certified dieticians/nutritionists. They instructed the participants on two methods of dietary recall. The first of them was to complete a food frequency questionnaire (FFQ) [27] following blood collection, which had been previously validated and utilized for an athlete population [28]. The participants completed the FFQ to recall the “frequency” of intake over the previous 8 weeks of 139 different portion sizes of foods and drinks.

Frequency categories were based on the number of times and portion sizes, which a food/drink was consumed per day, per week or per month. In addition, daily numbers of portions of different food groups were calculated. Daily consumption of energy (kcal) and each macronutrient and micronutrient were determined by dividing the reported intake by the frequency in days [27].

The second method was for athletes to complete a 7-day dietary recall at T1 and T2 of the previous 7 days to compare if these results were similar to that of the FFQ. If participants could weigh food, then this data was included in the dietary recall; however, if weighing food was not possible serving sizes consumed were estimated from the standard weight of food items or by determining portion size via looking at a book with 500 photographs of foods. Food values were then converted into intakes of total energy and micronutrients by a validated software package (Easy diet^©^, online version). This software package was developed by the Spanish Centre for Higher Studies in Nutrition and Dietetics (CESNID), which is based on Spanish tables of food composition [8].

### 2.5. Body Composition and Anthropometric Measures

Anthropometric measurements were taken following “The International Society for the Advancement of Kinanthropometry” (ISAK) protocol [29]. Additionally, the same internationally certified anthropometrist (ISAK level 3) took measurements for all participants. All measurements were undertaken in duplicate to establish within-day retest reliability. If the difference between the duplicate measures exceeded 5% for an individual skinfold, a third measurement was taken. The mean of duplicate or median of triplicate anthropometric measurements were used for all analysis. Height (cm) was measured using a SECA measuring rod, with a precision of 1 mm, while BM (kg.) was assessed by a SECA model scale, with a precision of 0.1 kg. Body mass index (BMI) was calculated using the formula BM height^2^ (kg/ m^2^). The sum of 4 skinfolds (mm) (Triceps, subscapular, suprailiac, and abdominal) was calculated, previously analyzed with a Harpenden^®^ skinfold caliber, with a precision of 0.2 mm. Absolute fat mass and muscle mass (absolute and percentage) were predicted using the Faulkner equation [30]. Lastly, absolute fat free mass was calculated as body mass (kg) − fat mass (kg).

### 2.6. Statistical Analysis

Data are presented as means and standard deviations. Analyses were performed using SPSS software version 24.0 (SPSS, Inc, Chicago, IL, USA). Statistical significance was indicated when *p* < 0.05.

Firstly, Kolmogorov–Smirnov tests were performed on the values of the parameters studied (*n* < 50) to decide parametric or non-parametric data. Secondly, the homoscedasticity of the variables analyzed by the Levene test was checked. As the distribution of all the parameters was normal, the results are presented as means ± standard deviation. Differences from T1 to T2 were assessed by a non-paired Student’s *t*-test. Likewise, an independent Student´s *t*-test was performed to calculate the differences in different parameters studied in CG and VD3G between baseline (T1) and after 8 weeks (T2).

Repeated-measures ANOVA with group factor (GC and GVD3) were used to examine the existence of an interaction effect of vitamin D3 treatment throughout the study (time × group) on hematological parameters, iron metabolism, and hormonal, that is, if the group-by-time interaction of the different values was the same or different between the groups throughout the study. Effect sizes among participants were calculated using partial eta square (η^2^p). Since this measure is likely to overestimate effect sizes, values were interpreted according to Ferguson [31] which indicates that there has been no effect if 0 ≤ η^2^p < 0.05; a minimum effect if 0.05 ≤ η^2^p < 0.26; a moderate effect if 0.26 ≤ η^2^p < 0.64; and a strong effect if η^2^p ≥ 0.64.

A McNemar test was performed to compare differences in vitamin D status between groups at baseline (T1) and after 8 weeks of treatment (T2). In order to compare the differences in vitamin D status, several groups were identified in which non-optimal level of 25(OH)D for athletic performance enhancement were considered below 50 ng/mL and within the optimal values between 50 and 100 ng/mL [3].

Finally, to analyze if 25(OH)D was a good predictor of testosterone, cortisol, and TT/C ratio, a stepwise regression model was performed using anabolic–catabolic hormones as the dependent variables and 25(OH)D serum level as predictors.

## 3. Results

In Figure 1, a statistically significant increase in the serum concentration of 25(OH)D in the VD3G can be observed (T1: 26.24 ± 8.18 ng/mL vs. T2: 48.12 ± 10.88 ng/mL, *p* < 0.001; η^2^p = 0.504), whereas in the CG no statistically significant changes were observed through the study (T1: 30.76 ± 6.95 ng/mL vs. T2: 35.14 ± 7.96 ng/mL; *p* = 0.056).

Table 1 shows the hematological and iron metabolism analytical values at (T1), as well as (T2) in both groups. Statistically significant differences in the group-by-time interaction of hemoglobin (*p* = 0.009; η^2^p = 0.354), hematocrit (*p* = 0.019; η^2^p = 0.300), and transferrin (*p* = 0.007; η^2^p = 0.374) between the two groups through the study were observed. Moreover, Table 2, presents in the CG a significant decrease (*p* < 0.05) in hemoglobin during the study (T1: 15.54 ± 0.88 vs. T2: 15.09 ± 0.82 ng/mL), and a significant increase (*p* < 0.05) in VD3G in transferrin levels over 8 weeks (T1: 254.22 ± 20.69 vs. 270.44 ± 20.08 mg/dL).

Table 2 describes the hormonal analytical values pre- and post-supplementation in both study groups. It can be observed that there were no statistically significant differences in the group-by-time interaction of the hormonal parameters between T1 and T2, and between the two supplementation groups for cortisol (*p* = 0.561; η^2^p = 0.022), testosterone (*p* = 0.852; η^2^p = 0.002), and the testosterone and cortisol index (TT/C) (*p* = 0.613; η^2^p = 0.016). Likewise, Table 2 shows in both groups a significant decrease (*p* < 0.05) both in cortisol and testosterone during study.

Regarding the status of 25(OH)D serum levels, although no differences were observed between groups in T1, a statistically significant differences (*p* < 0.001) between CG and VD3G were described during study (Table 3). Concretely, in both CG and VD3G in T1, all of the rowers (100%) had non-optimal states of 25(OH)D for athletic performance enhancement. After eight weeks of the study, the CG showed that all of the rowers (100%) maintained non-optimal serum 25(OH)D levels. However, in T2, VD3G presented that 50% of the rowers had an optimal state of serum 25(OH)D for athletic performance enhancement.

Table 4 shows the energy, macro, and micronutrient intake of the elite athletes during study. There was no significant difference between groups for total energy and micronutrients intake (*p* > 0.05), including vitamin D (CG: 16.93 ± 6.31 vs. VD3G: 15.34 ± 5.32 µg) and iron (CG: 16.25 ± 6.16 vs. VD3G: 16.65 ± 5.97 mg).

Table 5 shows the daily numbers of portions of different food groups of the elite athletes during study. There was no significant difference between groups (*p* > 0.05), including dietary vitamin D sources such as milk and dairy products (CG: 2.81 ± 0.35 vs. VD3G: 2.76 ± 0.32 servings/day) and fish (CG: 1.89 ± 0.33 vs. VD3G: 1.95 ± 0.41 servings/day).

Table 6, displays the body composition of the two groups at baseline and after eight weeks of treatment. Although in both groups all measures presented significant differences during the study (*p* < 0.05), the group-by-time interaction of them between groups did not show any significant difference (*p* > 0.05).

Table 7 shows a regression multivariate analysis with vitamin D as independent variable and catabolic–anabolic hormones (cortisol and testosterone) and its ratio as the dependent variables. 25(OH)D was found to be a significant predictor of these hormones. Specifically, 25(OH)D were significantly associated with cortisol (y = 24.796 + −0.147 × 25(OH)D) and testosterone (y = 6.425 + 0.44 × 25(OH)D). However, it did observe that 25(OH)D was a predictor of TT/C ratio.

## 4. Discussion

The main purpose of this intervention study was to analyze the effect of supplementation with 3000 IU/day of vitamin D3 during eight weeks on the hematological and iron metabolism profile, as well as on muscle recovery measured on anabolic–catabolic hormones, testosterone, and cortisol in elite traditional rowers. In general, the results revealed that this supplementation presented a statistically higher group-by-time interaction of the hemoglobin, hematocrit, and transferrin values, and an improvement of the serum levels of vitamin D in the supplemented athletes.

It is widely known that supplementation with doses of 1000–10,000 IU/day of vitamin D3 are effective to treat people with 25(OH)D deficiency [32]. There are several authors who have shown that a supplementation with 3000 IU/day adapts the levels of 25(OH)D in the general population [10,11]. In this way, the study also showed that athletes treated with 3000 IU/day of vitamin D3 improved 25(OH)D levels. Moreover, it was presented that all athletes treated with vitamin D3 showed adequate levels of 25(OH)D at the end of the treatment, confirming this relationship.

Traditional rowing is within the disciplines of strength-endurance and among the specialties of long-term endurance, where around 85–90% of the energy required in this type of sport is supplied via aerobic metabolism [33]. In this sense, a high correlation has been found between aerobic capacity and high vitamin D concentration, both naturally and as a result of supplementation [34]. Therefore, vitamin D level is associated with endurance performance [6].

More concretely, a high maximum oxygen consumption (VO_2max_) is the best indicator of the rower’s aerobic power [33]. In this line, classical physiological studies showed that one of the main limiting factors of the VO_2max_ is the transport of oxygen, which is a function in which hemoglobin actively participates [35]. On the other hand, the continuous practices and competitions in this type of sport have been related to a reduction of erythrocytes and the concentration of hemoglobin, and therefore, with the prevalence of iron deficiency [36,37], which could compromise performance in rowing practice. With vitamin D supplementation, a decrease in the production of cytokines will occur, which could lead to a decrease in hepcidin, thus improving the iron status [2], and thus, achieving an improvement or at least a maintenance in hematological and iron metabolism values. In that way, the study shows how vitamin D supplementation could prevent the decrease of hemoglobin, hematocrit, and transferrin parameters.

In the case of transferrin, iron transporter proteins are therefore also considered an indicator of iron status [17], and in this study, statistical differences were obtained between groups. In particular, GVD3 showed higher changes in transferrin levels throughout the study. Regarding this topic, Blanco-Rojo et al. [38] had already observed in young women with iron deficiency a condition in which there is a decrease in iron, a positive correlation between better levels of 25(OH)D and an increase in transferrin. Therefore, vitamin D supplementation could improve iron status by increasing transferrin values.

Regarding hemoglobin, which is responsible for oxygen transport, and therefore, with a crucial role in VO_2max_ values, in this research, significant differences were also observed throughout the study, with higher values of hemoglobin in the VD3G. Anyway, the effect of vitamin D3 supplementation (6000 IU/day vitamin D3 for eight weeks) on aerobic capacity expressed as VO_2max_ had already been demonstrated in 14 professional Polish rowers, who showed a significant increase in VO_2max_ and anaerobic threshold values [39]. For that reason, vitamin D supplementation could influence the VO_2max_ value given that CYP enzymes that activate vitamin D3 have proteins with heme groups, which could potentially increase the affinity of oxygen binding to hemoglobin [40].

On the other hand, Jakovic et al. [41] presented a positive linear correlation between hemoglobin and hematocrit, since this latter parameter refers to the volume of erythrocytes in the blood, and hemoglobin is a hemoprotein contained in these erythrocytes. Therefore, high hematocrit values would represent high VO_2max_ values. In this study, hematocrit values were better in T2 in both rowers that were supplemented with vitamin D. So, for the hormonal profile, an increase in testosterone concentration would lead to an improvement in endogenous muscle recovery, and in addition to greater muscle hypertrophy, an improvement in strength production [42,43].

In this line, several studies have observed a positive association between higher intakes of dietary vitamin D with higher testosterone levels [44] and higher levels of androgens [45]. However, Sim et al. [46] did not find a significant correlation between 25(OH)D levels and total testosterone in a cross-sectional study of 1559 Korean men. This controversy in the results can be due to the lack of similarity in the methodology of these studies, since none of the abovementioned supplemented with vitamin D were athletes. To the best of our knowledge, only one study has been found with athletes [22], where higher 25(OH)D levels have been associated with higher testosterone and lower cortisol levels in 45 soccer players. Thus, the results showed the same associations.

On the contrary, in this study, it was not obtained statistically significant results in relation to testosterone group-by-time interaction and vitamin D supplementation that could corroborate the initial hypothesis. The data was opposite to those obtained by Pilz et al. [23] who observed higher levels of testosterone in men who had been supplemented with 3332 IU/day of vitamin D during one year. While these men were not athletes, it seems that a larger dose for a longer time could be effective in achieving improvements in serum testosterone.

On the other hand, for the role of cortisol in the state of muscle recovery, a high cortisol value may be indicative of an inadequate muscle recovery [21]. In fact, vitamin D3 is a nuclear receptor ligand that shows similarities with other ligands such as corticosteroids (cortisol), the main effector molecules of the hypothalamic–pituitary–adrenal axis or stress. Therefore, an increase in vitamin D3 levels could produce a decrease in cortisol values and stress levels [47]. Regarding this parameter, although there were no statistically significant differences in the group-by-time interaction of their levels between both groups throughout the study, a trend of higher values was observed in the GVD3 than in the GC, indicating a greater reduction of stress throughout the study in GVD3 [48], which is beneficial for any athlete. These same results were obtained in a prospective study of 41 women with multiple sclerosis supplemented for 16 weeks with 4000 IU/day of vitamin D3, in which no statistically positive differences were found between 25(OH)D levels. However, cortisol levels were observed in the group supplemented with vitamin D3 [47]. Although the participants of this study were non-sports women, it seems that in this case, a higher dose for a longer time did not prove to be effective.

As limiting factors, it is important to highlight the fact of the small sample size; however, it is true that it is very difficult to obtain larger samples in elite sports. In addition, another bias could be to have performed the study from April to June, months in which the skin is exposed to adequate ultraviolet B radiation from the sun (UVB) producing an optimal formation of specific vitamin D3 isomers, and therefore, good levels of 25(OH)D [40]. However, it should be noted that all rowers (100.0%) started from a deficit state even though this was a time of year which makes vitamin D3 supplementation fully recommendable, and therefore, the realization of this study. In this line, Ogan and Pritchett [49] stated that the possibility of athletes requiring increased intake of vitamin D is due to the active use of vitamin D in many metabolic pathways.

On the other hand, the lack of significant results in certain hematological parameters, such as ferritin, may be due to the fact that, as is already known, these values decrease as the sports season progresses and as more volume of aerobic exercise is carried out [50]; however, the study’s aim is to cushion this decrease in values as demonstrated by transferrin. Another limitation would be the fact of grouping the rowers by their levels of 25(OH)D for the improvement of sports performance instead of the criteria used for the general population.

In contrast, as a strength of study, the diet ingested by the athletes was controlled as well as the body composition throughout the intervention process, so that these parameters did not influence the final results and could present errors in the effect of the vitamin D.

Future investigations should increase the dose and supplementation time in strength-endurance athletes, and finally, it is suggested analyzing the kinetics of vitamin D during the winter months and throughout a full season. In the same way, future research should be oriented to verify the effects of vitamin D supplementation on hematological and hormonal values based on previous 25(OH)D levels.

## 5. Conclusions

In summary, oral supplementation with 3000 IU/day of vitamin D3 for eight weeks was shown to be sufficient to avoid a decline on hematological levels (hemoglobin and hematocrit) as well as an increase on transferrin levels. In the same way, it was enough to improve levels of 25(OH)D. However, although 3000 IU/day of vitamin D3 over eight weeks was not enough to show higher group-by-time interaction of testosterone and cortisol levels in elite traditional rowers, it was observed that serum 25(OH)D levels were associated with anabolic–catabolic hormones. Likewise, although it was not sufficient to enhance muscle recovery observed by testosterone and cortisol responses, it was observed that serum 25(OH)D levels were a predictor of anabolic and catabolic hormones.

## Figures and Tables

**Figure 1 nutrients-10-01968-f001:**
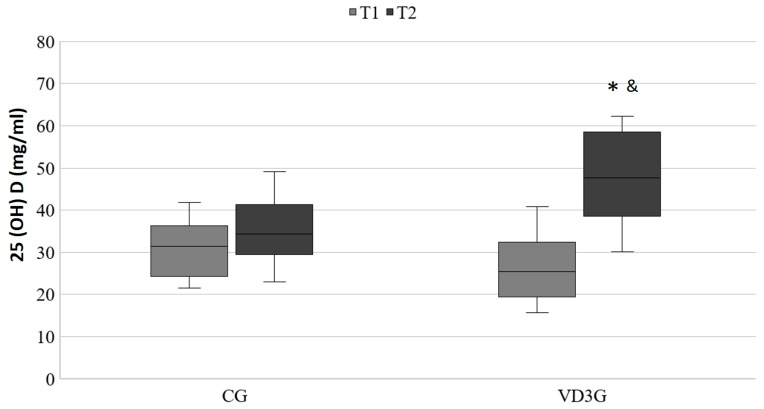
Serum 25(OH)D in the control group (CG) and vitamin D3 treatment group (VD3G) at baseline (T1) and after 8 weeks (T2). P: group-by-time interaction (*p* < 0.05 all such occurrences). Two-factor repeated-measures ANOVA. * Significantly different between phases (T1 vs. T2). *p* < 0.05. ^&^ Significantly different between groups (CG vs. VD3G). *p* < 0.05.

**Table 1 nutrients-10-01968-t001:** Hematological parameters in control group (CG) and vitamin D3 treatment group (VD3G) at baseline (T1) and after 8 weeks (T2).

	T1	T2	P	η^2^p
**Serum Iron (μg/dL)**				
CG	81.33 ± 17.75	72.78 ± 11.26	0.466	0.034
VD3G	76.67 ± 23.7	77.11 ± 26.51		
**Ferritin (ng/mL)**				
CG	74.67 ± 42.27	84.44 ± 28.05	0.403	0.044
VD3G	81.44 ± 38.26	84.33 ± 40.82		
**Hemoglobin (g/dL)**				
CG	15.54 ± 0.88	15.09 ± 0.82 *	0.009	0.354
VD3G	14.76 ± 0.77	14.76 ± 0.58		
**Hematocrit (%)**				
CG	46.04 ± 2.55	45.29 ± 2.12	0.019	0.300
VD3G	44.59 ± 2.27	45.07 ± 1.74		
**Transferrin (mg/dL)**				
CG	252.89 ± 46.51	253.22 ± 38.86	0.007	0.374
VD3G	254.22 ± 20.69	270.44 ± 20.08 *		

Data are expressed as mean ± standard deviation. P: group-by-time interaction (*p* < 0.05, all such occurrences). Two-factor repeated-measures ANOVA. * Significantly different between phases (T1 vs. T2), *p* < 0.05.

**Table 2 nutrients-10-01968-t002:** Hormonal parameters in the control group (CG) and vitamin D3 treatment group (VD3G) at baseline (T1) and after 8 weeks (T2).

	T1	T2	P	η^2^p
**Cortisol (µg/dL)**				
CG	20.44 ± 4.71	17.81 ± 2.16 *	0.561	0.022
VD3G	21.73 ± 4.61	18.18 ± 4.25 *		
**Testosterone (ng/mL)**				
CG	5.06 ± 1.41	4.37 ± 0.96 *	0.852	0.002
VD3G	5.37 ± 1.5	4.73 ± 1.28 *		
**Ratio Testosterone/Cortisol**				
CG	25.22 ± 6.18	24.49 ± 3.71	0.613	0.016
VD3G	25.66 ± 7.98	27.65 ± 9.53		

Data are expressed as mean ± standard deviation. P: group-by-time interaction (*p* < 0.05, all such occurrences). Two-factor repeated-measures ANOVA. * Significantly different between phases (T1 vs. T2), *p* < 0.05.

**Table 3 nutrients-10-01968-t003:** State of serum 25(OH)D in the control group (CG; *n* = 9) and vitamin D3 treatment group (ITG; *n* = 9) at baseline (T1) and after 8 weeks (T2).

	T1		T2		P
	**Non-Optimal**	**Optimal**	**Non-Optimal**	**Optimal**	<0.001
**CG**	18 (100%)	0 (0%)	18 (100%)	0 (0%)	
**VDG3**	18 (100%)	0 (0%)	9 (50%)	9 (50%)	

Data expressed in frequency (%). Non-optimal level of 25(OH)D for athletic performance enhancement: <50 ng/mL; optimal level of 25(OH)D for athletic performance enhancement: ≥50 ng/mL [3]. P: McNemar test.

**Table 4 nutrients-10-01968-t004:** Energy and micronutrients intake in control group (CG) and vitamin D3 group (VD3G) during 8 weeks of study.

	CG (*n* = 18)	VD3G (*n* = 18)	P
Energy (Kcal)	2818.9 ± 748.5	2942.4 ± 543.9	0.668
Proteins (g)	144.1 ± 37.1	128.3 ± 44.6	0.391
Fats (g)	138.3 ± 43.6	142.0 ± 43.9	0.848
Carbohydrates (g)	249.7 ± 118.0	286.1 ± 64.0	0.384
Ca (mg)	981.9 ± 628.7	806.9 ± 344.4	0.433
Mg (mg)	377.4 ± 139.8	391.7 ± 120.4	0.804
P (mg)	1704.8 ± 595.0	1734.1 ± 416.9	0.987
Fe (mg)	16.3 ± 6.2	16.7 ± 6.0	0.880
Zn (mg)	13.0 ± 3.9	13.7 ± 2.5	0.591
Vitamin A (µg)	959.2 ± 504.5	714.6 ± 374.2	0.219
Vitamin D (µg)	16.9 ± 6.3	15.3 ± 5.3	0.415
Vitamin E (mg)	20.8 ± 12.1	16.8 ± 7.4	0.369
Thiamin (mg)	1.4 ± 0.6	1.8 ± 0.6	0.148
Riboflavin (mg)	2.1 ± 0.75	2.0 ± 0.6	0.832
Niacin (mg)	34.3 ± 10.9	35.4 ± 10.5	0.812
Vitamin B6 (mg)	3.23 ± 0.94	2.9 ± 1.0	0.477
Folic Acid (mg)	428.2 ± 112.3	339.2 ± 108.4	0.081
Vitamin B12 (mg)	8.6 ± 3.7	8.1 ± 3.4	0.719
Vitamin C (mg)	179.7 ± 48.6	138.2 ± 88.4	0.206

Data are expressed as mean ± standard deviation. P: Significantly different between groups by independent *t*-test.

**Table 5 nutrients-10-01968-t005:** Daily number of portions of different food groups in the control group (CG) and vitamin D3 group (VD3G) during 8 weeks of study.

	CG (*n* = 18)	VD3G (*n* = 18)	P
Cereals/grains (150 g cooked weight)	4.26 ± 0.35	4.31 ± 0.42	0.7004
Milk and dairy products (200 ml)	2.81 ± 0.35	2.76 ± 0.32	0.6575
Fruits (120–150 g)	2.47 ± 0.82	2.51 ± 0.72	0.8773
Vegetables (120–150 g)	2.67 ± 0.65	2.47 ± 0.74	0.3950
Oil and fats (12 g)	2.28 ± 0.32	2.31 ± 0.41	0.8081
Legumes (150 g cooked weight)	0.51 ± 0.23	0.50 ± 0.21	0.8924
Dry fruits (20–30 g)	0.42 ± 0.18	0.38 ± 0.21	0.5436
Fish (120–150 g)	1.89 ± 0.33	1.95 ± 0.41	0.6317
Meat and meat products (120–150 g)	1.09 ± 0.45	1.00 ± 0.64	0.6286
Eggs (units)	0.50 ± 0.01	0.50 ± 0.01	1.000
Sugar and sweets (30–50 g)	0.87 ± 0.25	0.77 ± 0.18	0.1774
Nonalcoholic beverages (200 ml)	0.27 ± 0.18	0.30 ± 0.15	0.5905

Data are expressed as mean ± standard deviation. P: Significantly different between groups by independent *t*-test.

**Table 6 nutrients-10-01968-t006:** Body composition in the control group (CG) and vitamin D3 group (VD3G) at baseline (T1) and after 8 weeks (T2).

	T1	T2	P	η^2^p
**Body Mass (kg)**				
GC	77.0 ± 7.6	76.3 ± 7.3 *	0.817	0.005
GVD3	76.2 ± 10.1	76.3 ± 11.0 *		
**BMI**				
GC	23.46 ± 1.63	23.25 ± 1.58 *	0.777	0.008
GVD3	23.50 ± 1.35	23.51 ± 1.59		
**Sum 4 Skinfolds (mm)**				
GC	47.15 ± 11.35	44.10 ± 8.84 *	0.253	0.117
GVD3	49.59 ± 18.58	41.83 ± 11.77		
**Fat Mass (%)**				
GC	13.00 ± 1.74	12.53 ± 1.35 *	0.253	0.117
GVD3	13.37 ± 2.84	12.18 ± 1.80 *		
**Fat Mass (kg)**				
GC	10.02 ± 1.82	9.56 ± 1.37 *	0.507	0.041
GVD3	10.25 ± 2.63	9.38 ± 2.29 *		
**Free Fat Mass (kg)**				
GC	67.00 ± 6.54	70.85 ± 10.94 *	0.348	0.080
GVD3	65.94 ± 8.61	66.57 ± 9.19 *		

Data are expressed as mean ± standard deviation. P: group-by-time interaction (*p* < 0.05. all such occurrences). Two-factor repeated-measures ANOVA. * Significantly different between phases (T1 vs. T2), *p* < 0.05.

**Table 7 nutrients-10-01968-t007:** Regression multivariate analysis with catabolic–anabolic hormones as the dependent variable and 25(OH) D as predictor.

	Unstandardized Coefficients		Standardized Coefficients	t	P
	B	Std. Error	Beta		
**Cortisol**					
(Constant)	24.796	2.089		11.82	<0.000
25(OH)D	−0.147	0.057	−0.407	−2.597	0.014
**Testosterone**					
(Constant)	6.425	0.649		9.902	<0.000
25(OH)D	0.44	0.018	0.394	2.501	0.017
**Testosterone/cortisol Ratio**					
(Constant)	26.755	3.772		7.092	<0.000
25(OH)D	−0.029	0.102	−0.48	−0.279	0.782

*p* < 0.05 indicates significative association between predictor (25(OH) D) and dependent variables (catabolic–anabolic hormones).

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
