# Peer review of "Effects of Vitamin D Supplementation on Haematological Values and Muscle Recovery in Elite Male Traditional Rowers"

_nutrients, 2018, doi:10.3390/nu10121968_

Round 1
Reviewer 1 Report
The manuscript entitled “Effects of vitamin D supplementation on haematological values and muscle recovery in elite male traditional rowers” presents interesting issue, but some important corrections are necessary.
The study presents a major inconsistencies associated with a lot of issues. However, the most disturbing is the inconsistency associated with vitamin D – the most important factor analysed in the study.
Authors stated, that they planned the diet to meet the RDA levels of Institute of Medicine (IoM, 2006) (lines 116-118). But for the indicated reference, there is no RDA vitamin D recommendation of IoM, but only AI. There are new recommendations of IoM for vitamin D (RDA of 15 ug) but it is the other reference (Institute of Medicine; Committee to Review Dietary Reference Intakes for Vitamin D and Calcium; Food and Nutrition Board. Dietary Reference Intakes for Calcium and Vitamin D; National Academies Press: Washington, DC, USA, 2011). However, how is it possible, that Author planned a diet to meet the recommendation of 15 ug, but the real vitamin D intake for a groups was 6.9 or 5.3 (Table 2)?
There are two possibilities:
- Authors planned a diet on a level below the currently recommended (5 ug instead of 15 ug) – so they planned a deficiency
- Authors planned a diet on a recommended level but the diet was improperly followed (so the real observed intake was significantly lower) – so they planned proper diet but insufficient intake occurred
In both cases the diet was improperly balanced and Authors in fact assessed the effect of vitamin D supplementation in the case of deficiency ones. It is confirmed by a high frequency of deficiency in a groups (Table 1). It does not correspond the aim of the study, as it was not specified, that Authors planned to generate a deficiency of assessed nutrient.
In the case of a group with a generated deficiency the observed conclusions may have been supposed before conducting study – in general, the supplementation of a nutrient in the case of deficiency will cause solving the problem of deficiency.
However, the influence on the iron metabolism in the disturbed conditions of generated deficiency should not be assessed (it was not the aim of the study to generate a deficiency).
General:
The manuscript is in general shabbily prepared (typestyle – e.g. e-mail addresses, lack of formatting according to recommendations – e.g. references, numbers of chapters, highlighting in colour – e.g. keywords)
It seems, that Authors are not native English speakers (lack of English words but using Spanish ones – e.g. “hidroxivitamina”, improper verbs – e.g. “levels was” – line 38) – the manuscript should be corrected by native English speaker or preferably by the professional agency.
The applied abbreviation for vitamin D (VITD) is rather unusual – it should be changed or preferably this abbreviation should be removed
Authors should avoid personal description (e.g. “our study”) and instead use less personal phrases (e.g. “the study”).
Abstract:
Authors should not present very basic or even trivial information (that 25(OH)D is metabolite of vitamin D, that supplementation may prevent deficiency)
It should be precisely justified why testosterone and cortisol were assessed
Lines 30-36 – Authors should precisely indicate which analysis were done – was it comparison between groups or comparison between T1 and T2 values
“Behaviour” – should be explained – which biochemical parameters were assessed
Lines 41-42 - testosterone and cortisol do not allow to conclude about enhanced muscle recovery
Introduction:
The section requires a really major corrections – Authors should properly justify the study, while they should present precisely the specific information with no generalizing and no basic information presented. Moreover, the section should not present all the information that Authors have but only the most important ones, in order to properly justify the study.
Authors tend to present very basic or even trivial information as a novelty (e.g. “Judd and Tangpricha concluded a significant association between low levels of 25-hidroxivitamina D (25(OH)D) (….), and the risk of suffering from different types of diseases” – it is well known fact for ages, not concluded only by Judd and Tangpricha but by a number of authors) – such information should be removed as the readers of Nutrients are in general familiar with the role of vitamin D and only more specific information should be presented.
Authors tend to generalize the observations from the referred studies – e.g. they refer the article about cardiovascular diseases, but conclude about “different types of diseases”, they refer the article about soccer, but conclude about “sports sciences”, etc.
Authors should precisely indicate information – e.g. they indicated “low availability of VITD in some periods of the year”, but it is not “low availability” but rather low synthesis.
Lines 59-62 – are not related to the aim of the study – Authors present information about doses higher than 4000 IU, while they in the fact applied the dose of 3000 IU
In the Introduction section, Authors totally ignored the role of diet as a source of vitamin D – it can not be omitted.
Lines 95-102 – Authors should precisely present the aim of the study, in a one sentence (e.g. “the aim of the study was…”) instead of presenting what was done and specifying hypothesis.
Methods:
The section should be rather “Materials and Methods”
Lines 108-109 – it should be specified – how long were the variables controlled – during the whole study?
Line 114 – what do Authors mean by “prescribed” – did each rower individually prepare his meals or was it prepared by qualified workers?
Authors should precisely indicate what was done – if “T2” means “post-treatment”, what does mean “the last day of T2”?
Number of repetitions for each measurement should be indicated.
FFQ – was only the frequency assessed, or the serving sizes were also assessed?
If the distribution is normal, the mean values should be presented (accompanied by SD), but if it is different than normal, the median, accompanied by minimum and maximum values should be presented – it should be specified that distribution is normal if it is.
Results:
The section must be corrected accordingly, taking into account the indicated above problems
Authors must correct the conducted analysis by specifying what exactly was the aim – was it to assess the influence of supplementation in (a) vitamin D deficiency group, (b) group with no vitamin D deficiency, (c) sub-group characterized by insufficient intake of vitamin D, (d) sub-group characterized by sufficient intake of vitamin D?
The separate analysis in sub-groups are highly recommended.
Discussion:
The section must be corrected accordingly, taking into account the indicated above problems
The more insightful discussion is highly recommended. Authors must specify what exactly was the aim – was it to assess the influence of supplementation in (a) vitamin D deficiency group, (b) group with no vitamin D deficiency, (c) sub-group characterized by insufficient intake of vitamin D, (d) sub-group characterized by sufficient intake of vitamin D? Afterwards, they must precisely discuss the mechanism for a specific sub-group.
The limitations paragraph should be corrected accordingly.
Conclusions:
The section must be corrected accordingly, taking into account the indicated above problems
References:
Authors should precisely indicate the references – e.g. for the reference no. 31, they specified pp. 543 (Otten, J.J.; Hellwig, J.P.; Meyers, L.D.; Institute of Medicine. Dietary Reference Intakes: The Essential Guide to Nutrient Requeriments.; The National Academies Press: Washington, D.C., 2006; pp. 543.) – but in the book, page 543 is blank – with no information
Author Response
Dear Reviewer,
We appreciate again the time you devoted to reading our manuscript and helping us to craft an improved version. We are pleased to clarify your concern which we believe will improve the impact and quality of your work. Please find below our response to your observation. We have made a concerted attempt to systematically address the specific concerns raised for this revision and we have highlighted the alterations to this revision within the manuscript in green for your convenience.
Reviewer(s)' Comments to Author:
Reviewer 1:
Comments and Suggestions for Authors
The manuscript entitled “Effects of vitamin D supplementation on haematological values and muscle recovery in elite male traditional rowers” presents interesting issue, but some important corrections are necessary.
The study presents a major inconsistencies associated with a lot of issues. However, the most disturbing is the inconsistency associated with vitamin D – the most important factor analysed in the study.
Reviewer 1: Authors stated, that they planned the diet to meet the RDA levels of Institute of Medicine (IoM, 2006) (lines 116-118). But for the indicated reference, there is no RDA vitamin D recommendation of IoM, but only AI. There are new recommendations of IoM for vitamin D (RDA of 15 ug) but it is the other reference (Institute of Medicine; Committee to Review Dietary Reference Intakes for Vitamin D and Calcium; Food and Nutrition Board. Dietary Reference Intakes for Calcium and Vitamin D; National Academies Press: Washington, DC, USA, 2011). However, how is it possible, that Author planned a diet to meet the recommendation of 15 ug, but the real vitamin D intake for a groups was 6.9 or 5.3 (Table 2)?
There are two possibilities:
- Authors planned a diet on a level below the currently recommended (5 ug instead of 15 ug) – so they planned a deficiency
- Authors planned a diet on a recommended level but the diet was improperly followed (so the real observed intake was significantly lower) – so they planned proper diet but insufficient intake occurred
In both cases the diet was improperly balanced and Authors in fact assessed the effect of vitamin D supplementation in the case of deficiency ones. It is confirmed by a high frequency of deficiency in a groups (Table 1). It does not correspond the aim of the study, as it was not specified, that Authors planned to generate a deficiency of assessed nutrient.
In the case of a group with a generated deficiency the observed conclusions may have been supposed before conducting study – in general, the supplementation of a nutrient in the case of deficiency will cause solving the problem of deficiency.
However, the influence on the iron metabolism in the disturbed conditions of generated deficiency should not be assessed (it was not the aim of the study to generate a deficiency).
Answer: Thank you for your big observation. It was indeed a type when transcribing the vitamin D data into the table 2. Instead of 16.9±6.3 and 15.3±5.3 we transcribe by mistake 6.9±6.3 and 5.3±5.3 respectively. Of course, it has been something that has caused us some frustration because, as the reviewer says, vitamin D is the core of study of this manuscript.
Reviewer 1: The manuscript is in general shabbily prepared (typestyle – e.g. e-mail addresses, lack of formatting according to recommendations – e.g. references, numbers of chapters, highlighting in colour – e.g. keywords)
Answer: Thank you for your appreciation. The authors have adapted the manuscript to the standards of the journal
Reviewer 1: It seems, that Authors are not native English speakers (lack of English words but using Spanish ones – e.g. “hidroxivitamina”, improper verbs – e.g. “levels was” – line 38) – the manuscript should be corrected by native English speaker or preferably by the professional agency.
Answer: Thank you for your comment. Manuscript has been corrected by English speaker.
Reviewer 1: The applied abbreviation for vitamin D (VITD) is rather unusual – it should be changed or preferably this abbreviation should be removed
Answer:Thank you for your recommendation. Authors have changed VITD abbreviation for vitamin D.
Reviewer 1: Authors should avoid personal description (e.g. “our study”) and instead use less personal phrases (e.g. “the study”).
Answer:Thank you for this appreciation. Following reviewer consideration authors have changed our study for the study.
Abstract:
Reviewer 1: Authors should not present very basic or even trivial information (that 25(OH)D is metabolite of vitamin D, that supplementation may prevent deficiency).
Answer: Thank you for your observation. Considering that abstract should be no more extensive, authors have added this information in introduction section. The new paragraph is: “Deficient levels of 25-hydroxyvitamin D (25(OH)D) (<30 ng/ml),may compromise health and athletic performance. The supplementation with oral Vitamin D can favor the state of the iron metabolism and testosterone and cortisol as indicator of muscular recovery of the athlete.”.
Reviewer 1: It should be precisely justified why testosterone and cortisol were assessed
Answer:Thank you for your appreciation. Authors have included testosterone and cortisol in abstract´s introduction. “The supplementation with oral Vitamin D can favor the state of the iron metabolism and testosterone and cortisol as indicator of muscle recovery of the athlete.”.
Reviewer 1: Lines 30-36 – Authors should precisely indicate which analysis were done – was it comparison between groups or comparison between T1 and T2 values
Answer:Thank you for your observation. Although the authors analyzed the parameters with different test, in the abstract, we have included next paragraph: “Repeated-measures ANOVA with group factor (GC and GVD3) were used to examine if the interaction (behavior) of the different values was the same or different between the groups throughout the study (time x group) after vitamin D3 treatment. To analyze if 25(OH)D was a good predictor of testosterone, cortisol and TT/C ratio a stepwise regression model was performed.”.
Reviewer 1: “Behaviour” – should be explained – which biochemical parameters were assessed
Answer: Thank you for your interest. As the authors explain in the previous commentary Repeated-measures ANOVA with group factor (GC and GVD3) were used to examine if the interaction (behavior) of the different values was the same or different between the groups throughout the study (time x group) after vitamin D3 treatment
Reviewer 1: Lines 41-42 - testosterone and cortisol do not allow to conclude about enhanced muscle recovery
Answer:Thank you for your commentary. In this line, authors explained “However, although it was no sufficient to enhance muscle recovery observed by testosterone and cortisol responses, it observed that serum 25(OH)D levels were associated with anabolic and catabolic hormones”
Introduction:
Reviewer 1: The section requires a really major corrections – Authors should properly justify the study, while they should present precisely the specific information with no generalizing and no basic information presented. Moreover, the section should not present all the information that Authors have but only the most important ones, in order to properly justify the study.
Answer: Thank you for your recommendation. Authors have done reviewer’s recommendations. The authors believe that now the introduction is much better, thank you
Reviewer 1: Authors tend to present very basic or even trivial information as a novelty (e.g. “Judd and Tangpricha concluded a significant association between low levels of 25-hidroxivitamina D (25(OH)D) (….), and the risk of suffering from different types of diseases” – it is well known fact for ages, not concluded only by Judd and Tangpricha but by a number of authors) – such information should be removed as the readers of Nutrients are in general familiar with the role of vitamin D and only more specific information should be presented.
Answer: Thank you for your recommendation. Following this recommendation authors have deleted the next sentence “Judd and Tangpricha concluded a significant association between low levels of 25- hydroxy vitamin D (25(OH)D) (prohormone and metabolite of vitamin D), and the risk of suffering from different types of diseases [1].”
Reviewer 1: Authors tend to generalize the observations from the referred studies – e.g. they refer the article about cardiovascular diseases, but conclude about “different types of diseases”, they refer the article about soccer, but conclude about “sports sciences”, etc.
Answer:Thank you for your comment. In order to avoid generalizing and misunderstandings, the authors have eliminated the sentence, as indicated in the previous comment.
Reviewer 1: Authors should precisely indicate information – e.g. they indicated “low availability of VITD in some periods of the year”, but it is not “low availability” but rather low synthesis.
Answer: Thank you for your comment. Indeed, we cannot speak of availability but of synthesis. therefore, the authors have changed the word bioavailability to synthesis.
Reviewer 1: Lines 59-62 – are not related to the aim of the study – Authors present information about doses higher than 4000 IU, while they in the fact applied the dose of 3000 IU
Answer:Thank you for your recommendation. Following reviewer recommendation and given that lines 59-62 are not related to the aim of the study, authors have deleted these sentences “the European Food Safety Authority (EFSA) has set maximum tolerable limits for vitamin D supplementation of 4,000 IU/day for persons over 18 years [2]. However, the efficacy of supplementation with doses between 1,000-10,000 IU/day of vitamin D in people with a deficit of 25(OH)D (<30 ng/ml) has been demonstrated in different studies. [3]. In addition,”
Reviewer 1: In the Introduction section, Authors totally ignored the role of diet as a source of vitamin D – it can not be omitted.
Answer: Thank you for your observation. In this way, authors have added this sentence to avoid that source of vitamin D being omitted: Vitamin D is found naturally in only a few foods—fatty fish (i.e., mackerel, salmon, sardines, tuna), egg yolks, certain mushrooms—and in dairy products, margarine, ready-to-eat cereals, and fruit juices that have been fortified [9].
Reviewer 1: Lines 95-102 – Authors should precisely present the aim of the study, in a one sentence (e.g. “the aim of the study was…”) instead of presenting what was done and specifying hypothesis.
Answer: Thank you for your recommendation. Authors have adjusted the aim paragraph as: “Therefore, the main aim was to evaluate the influence of 8 weeks of supplementation with 3,000 IU/day of vitamin D3 on the hematological and iron metabolism profile, as well as in the analytical values of testosterone and cortisol in elite traditional rowers. In addition, a secondary aim was to examine the association between vitamin D and testosterone and cortisol.”
Methods:
Reviewer 1: The section should be rather “Materials and Methods”
Answer:Thank you for your recommendation. Authors have added Material and Methods in that section
Reviewer 1: Lines 108-109 – it should be specified – how long were the variables controlled – during the whole study?
Answer: Thank you for your observation. Authors have added “during the whole study” in the sentence. “The solar irradiance and outdoor air temperature were recorded every 10 min during the whole study by wireless Vantage Pro2™ Plus (Davis Instruments, Hayward, California, USA).”
Reviewer 1: Line 114 – what do Authors mean by “prescribed” – did each rower individually prepare his meals or was it prepared by qualified workers?
Answer: Thank you for your comment. the authors have changed prescribed by elaborated to avoid misunderstandings.
Reviewer 1: Authors should precisely indicate what was done – if “T2” means “post-treatment”, what does mean “the last day of T2”?
Answer: Thank you for your comment. Authors have included 2. Post-treatment (T2 - the day after 8 weeks of treatment” to avoid misunderstandings
Reviewer 1: Number of repetitions for each measurement should be indicated.
Answer: Thank you for your recommendation. Authors have added next sentences: “All measurements were undertaken in duplicate to establish within-day retest reliability. If the difference between the duplicate measures exceeded 5% for an individual skinfold, a third measurement was taken. The mean of duplicate or median of triplicate anthropometric measurements were used for all analysis.”.
Reviewer 1: FFQ – was only the frequency assessed, or the serving sizes were also assessed?
Answer: Thank you for your comment. This FFQ included portion sizes in each food and beverage. In this line, authors have included this data into paragraph: The FFQ, asked the participants to recall the ‘frequency’ of intake over the previous 8 weeks of 139 different portion sizes of foods and drinks. Frequency categories were based on the number of times and portion sizes, which a food/drink was consumed per day, per week or per month.”
Reviewer 1: If the distribution is normal, the mean values should be presented (accompanied by SD), but if it is different than normal, the median, accompanied by minimum and maximum values should be presented – it should be specified that distribution is normal if it is.
Answer: Thank you for your comment. Taking into to account that data had normal distribution, authors have deleted Mann-Whitney U test and Wilcoxon test in the Statistical Analysis section
Results:
Reviewer 1: The section must be corrected accordingly, taking into account the indicated above problems.
Answer: Thank you for your commentary. Authors have improved the results section taking into account the above problems.
Reviewer 1: Authors must correct the conducted analysis by specifying what exactly was the aim – was it to assess the influence of supplementation in (a) vitamin D deficiency group, (b) group with no vitamin D deficiency, (c) sub-group characterized by insufficient intake of vitamin D, (d) sub-group characterized by sufficient intake of vitamin D?
Answer: Thanks for your appreciation. This comment has made us think that taking into account that the maximum sport performance is obtained from 50 ng/ml, this should be the criterion by which the table 1 is guided. Thus, we have considered optimal levels for the maximum performance equal to or greater than 50 ng / ml, while we have considered levels not optimal for those below 50 ng / ml. Thus table 1 has remained as shown below. Also, the text referring to table 1 has been as follows.
Regarding the status of 25(OH)D serum levels, although no differences were observed between groups in T1, a statistically significant differences (p < 0.001) between CG and VD3G were observed during study (table 1). Concretely, in both CG and VD3G in T1, all of rowers (100%) had non-optimal state of 25 (OH) D for athletic performance enhancement. After 8 weeks of the study the CG showed that all of rowers (100%) maintained non-optimal serum 25(OH)D levels. However, in T2 VD3G presented that 50% of rowers had optimal State of serum 25(OH)D for athletic performance enhancement.
Table 1. State of serum 25(OH)D in control group (CG; n = 9) and vitamin D3 treatment group (ITG; n = 9) at baseline (T1) and after 8 weeks (T2).
T1 | T2 | P | |||
Non-Optimal | Optimal | Non-Optimal | Optimal | <0.001< span=""> | |
CG | 18 (100%) | 0 (0%) | 18 (100%) | 0 (0%) | |
VDG3 | 18 (100%) | 0(0%) | 9 (50%) | 9 (50%) | |
Data expressed in frequency (%). Non-Optimal level of 25(OH)D for athletic performance enhancement:<50 ng/ml; Optimal level of 25(OH)D for athletic performance enhancement: ≥ 50 ng/ml [13]. P: McNemar test.
Reviewer 1: The separate analysis in sub-groups are highly recommended.
Answer: Thank you for your recommendation. However, the new criteria used to classify the levels of 25 (OH) D means that in T1 all the rowers are classified as non-optimal, not being able separate analysis in sub-groups.
Discussion:
Reviewer 1: The section must be corrected accordingly, taking into account the indicated above problems
The more insightful discussion is highly recommended. Authors must specify what exactly was the aim – was it to assess the influence of supplementation in (a) vitamin D deficiency group, (b) group with no vitamin D deficiency, (c) sub-group characterized by insufficient intake of vitamin D, (d) sub-group characterized by sufficient intake of vitamin D? Afterwards, they must precisely discuss the mechanism for a specific sub-group.
Answer:Thanks for your recommendation. However, taking into account that finally it was decided to group the rowers according to the levels to improve sports performance and all the participants were in the same conditions at baseline, the authors consider that it is not necessary to separate the discussion proposed by the reviewer.
Reviewer 1: The limitations paragraph should be corrected accordingly.
Answer:Thank you for your recommendation. Authors have added next sentence in limitation section: “Another limitation would be the fact of grouping the rowers by their levels of 25 (OH) D for the improvement of sports performance instead of the criteria used for the general population.”
Likewise, authors have included next sentence in future investigations section: “In the same way, future researches should be oriented to verify the effects of vitamin D supplementation on hematological and hormonal values based on previous 25 (OH) D levels.”
Conclusions:
Reviewer 1: The section must be corrected accordingly, taking into account the indicated above problems
Answer: Thanks for your recommendation. Taking into account that the study has 2 objectives as we have already mentioned in previous answers, the authors have changed the conclusions: “In Summary, oral supplementation with 3000 IU/day of vitamin D3 for 8 weeks was shown to be sufficient to avoid a decline on hematological levels (hemoglobin and hematocrit) as well as an increase on transferrin levels. In the same way, it was enough to improve levels of 25(OH)D. However, although 3000 IU/day of vitamin D3 during 8 weeks was not enough to show higher behavior of testosterone and cortisol levels in elite traditional rowers, it observed that serum 25(OH)D levels were associated with anabolic-catabolic hormones. Likewise, although it was not sufficient to enhance muscle recovery observed by testosterone and cortisol responses, it observed that serum 25(OH)D levels were associated with anabolic and catabolic hormones.”.
References:
Reviewer 1: Authors should precisely indicate the references – e.g. for the reference no. 31, they specified pp. 543 (Otten, J.J.; Hellwig, J.P.; Meyers, L.D.; Institute of Medicine. Dietary Reference Intakes: The Essential Guide to Nutrient Requeriments.; The National Academies Press: Washington, D.C., 2006; pp. 543.) – but in the book, page 543 is blank – with no information
Answer: Thank you for your observation. Authors have deleted number of page of this book.

Reviewer 2 Report
Thank you for allowing me to review this manuscript. I found this to be an interesting topic and a well-written manuscript that is highly relevant at this time of year. The major comment relates to the lack of a rationale for the multiple regression. I am not sure what this adds to the manuscript.
As a consequence I only have a few minor suggestions / comments.
Title: Is the word 'traditional' required?
Abstract:
Line 20: A definition of 'low' would be a useful addition/
Line 26: Provide age as an integer. This also applies to the methods.
Line 30: Provide direction of differences, and through the results.
Line 41: Replace 'no sufficient' with 'not sufficient'
Introduction:
The opening paragraph would benefit by the addition of definitions regarding the values / cut offs for optimal sub-optimal vitamin D status - this could also highlight the inconsistencies in terminology.
Lines 67-85: Consider combining these two paragraphs and introduce the role of vitamin D before discussing iron metabolism. It is then unclear how recovery and cortisol fit into this. Consequently, there needs to be a stronger rationale for these components of the investigation.
Methods:
Consider including tables 1 and 2 in the methods as these provide useful background information regarding the participant's characteristics.
Lines 176-77: This sentence does not make sense, it appear that there could be a word missing.
Results:
Include the variables on the Y axis
Do not need to include the statistical results above the figure as these are presented in the text.
Unclear what table 4 adds to the manuscript
Discussion:
This section highlights the value of the study and provides some mechanistic insights. One minor point would be to be more specific than 'better' e.g. higher.
Author Response
Dear Reviewer,
We appreciate again the time you devoted to reading our manuscript and helping us to craft an improved version. We are pleased to clarify your concern which we believe will improve the impact and quality of your work. Please find below our response to your observation. We have made a concerted attempt to systematically address the specific concerns raised for this revision and we have highlighted the alterations to this revision within the manuscript in green for your convenience.
Reviewer(s)' Comments to Author:
Reviewer 2:
Reviewer 2: Thank you for allowing me to review this manuscript. I found this to be an interesting topic and a well-written manuscript that is highly relevant at this time of year. The major comment relates to the lack of a rationale for the multiple regression. I am not sure what this adds to the manuscript.
Reviewer 2: As a consequence I only have a few minor suggestions / comments.
Author’s response: Thanks so much. These minor suggestions have been changed.
Reviewer 2: Title: Is the word 'traditional' required?
Author’s response: Yes, this is the adequate term, given that in rowing world, we have two different modalities (Olympic rowing and this second option: a sport based on traditional activity in Basque Country). See this official website (https://es.wikipedia.org/wiki/Remo_(deporte)#Banco_fijo )
Reviewer 2: Line 20: A definition of 'low' would be a useful addition/
Author’s response: Thank you for your recommendation. To avoid misunderstandings, the authors have changed the word “low” by “deficient” levels (<30 ng/ml).
Reviewer 2: Line 26: Provide age as an integer. This also applies to the methods.
Author’s response: Thank you for your recommendation. Following reviewer recommendation authors have provided ages as an integer.
Reviewer 2: Line 30: Provide direction of differences, and through the results.
Author’s response: Thank you for your suggestion. Authors have changed this sentence by “Statistically significant differences increase were observed in the behavior of 25(OH)D in VD3G respect to CG during the study (p<0.001; VD3G (T1: 26.24 ± 8.18 ng/ml vs. T2: 48.12 ± 10.88 ng/ml) vs CG (T1: 30.76 ± 6.95 ng/ml vs. T2: 35.14 ± 7.96 ng/ml).”.
Reviewer 2: Line 41: Replace 'no sufficient' with 'not sufficient'
Author’s response: Good point. We have CHANGED it.
Reviewer 2: Introduction:
Reviewer 2: The opening paragraph would benefit by the addition of definitions regarding the values / cut offs for optimal sub-optimal vitamin D status - this could also highlight the inconsistencies in terminology.
Author’s response: Thanks for your suggestion. In this sense, in the second paragraph it is indicated which are the most adequate levels for the general population (30-100 ng / ml), being understood that everything that is below is a deficient level. In the same way, the optional level of 25 (OH) D is included for sports performance, this being 50 ng / ml or more.
Reviewer 2: Lines 67-85: Consider combining these two paragraphs and introduce the role of vitamin D before discussing iron metabolism. It is then unclear how recovery and cortisol fit into this. Consequently, there needs to be a stronger rationale for these components of the investigation.
Author’s response: Thanks for your recommendation. The authors have combined the second and third paragraphs. In addition, we have included the role of vitamin D in iron metabolism in the first place.
Reviewer 2: Methods:
Reviewer 2: Consider including tables 1 and 2 in the methods as these provide useful background information regarding the participant's characteristics.
Author’s response: Thanks for your suggestion. However, taking into account that tables 1 and 2 are the tables where statistics are performed, the authors agree that they should be maintained in the results section since they would lose their informative value. Anyway, if the reviewer insists, we would have no problem doing so.
Reviewer 2: Lines 176-77: This sentence does not make sense, it appear that there could be a word missing.
Author’s response: Ok, Agree. We have rewritten the sentence “The participants completed the FFQ to recall the ‘frequency’ of intake”.
Reviewer 2: Results:
Reviewer 2: Include the variables on the Y axis
Author’s response: Thank you for your recommendation. Authors have included the variables on the Y axis.
Reviewer 2: Do not need to include the statistical results above the figure as these are presented in the text.
Author’s response: Thank you for your recommendation. Authors have deleted the statistical results above the figures.
Reviewer 2: Unclear what table 4 adds to the manuscript
Author’s response: We appreciate your proposal, but as the reviewer observes we have added a second objective that obliges us to keep table 4 since it is the one that explains this objective.
Reviewer 2: Discussion:
Reviewer 2: This section highlights the value of the study and provides some mechanistic insights. One minor point would be to be more specific than 'better' e.g. higher.
Author’s response: Thanks for this detail. We have changed it thought the text.

Round 2
Reviewer 1 Report
The manuscript entitled “Effects of vitamin D supplementation on haematological values and muscle recovery in elite male traditional rowers” presents interesting issue, but some important corrections are necessary.
Major:
1. Authors explained their errors with the vitamin D content in the previous version of the manuscript and corrected them. But it is still not known what was the vitamin D level in the diet at baseline (in the diet prior to the study) – as patients were stated to be vitamin D deficient (Table 1) – it should be explained why.
2. Moreover, the intake of food products for 8 weeks of assessment should be presented, as Authors stated that the vitamin D intake was 15-16 ug (Table 2) – it should be presented what was the intake of specific food product groups (fish, meat, milk, dairy beverages, etc.)
General:
The manuscript is in general shabbily prepared (lack of formatting according to recommendations – e.g. references, or abstract, typos – e.g. “descrrbed”)
It seems, that Authors are not native English speakers (e.g. “descrrbed”) – the manuscript should be corrected by native English speaker or preferably by the professional agency.
The applied abbreviation for vitamin D (VITD) is rather unusual – it should be changed or preferably this abbreviation should be removed
Authors should avoid personal description (e.g. “our research”) and instead use less personal phrases (e.g. “the research”).
Abstract:
Authors should precisely indicate the information (e.g. “The supplementation with oral Vitamin D can favor the state of the iron metabolism and testosterone and cortisol as indicator of muscle recovery of the athlete.” – for all athletes or just deficiency ones?)
“Behaviour” – should be removed
Lines 45-46 - testosterone and cortisol do not allow to conclude about enhanced muscle recovery – the conclusion should not be associated with the possibility of such relation
Introduction:
The section still requires a really major corrections – Authors should properly justify the study, while they should present precisely the specific information with no generalizing and no basic information presented. Moreover, the section should not present all the information that Authors have but only the most important ones, in order to properly justify the study.
Authors tend to present very basic or even trivial information as a novelty (e.g. “Judd and Tangpricha concluded a significant association among deficient levels of 25-hydroxyvitamin D (…), and the risk of suffering from different types of diseases” – it is well known fact for ages, not concluded only by Judd and Tangpricha but by a number of authors) – such information should be removed as the readers of Nutrients are in general familiar with the role of vitamin D and only more specific information should be presented.
Authors tend to generalize the observations from the referred studies – e.g. they refer the article about cardiovascular diseases, but conclude about “different types of diseases”, etc.
In the Introduction section, Authors improperly presented a diet as a source of vitamin D – Authors referred not the databases of nutritional value of products, but some not very prominent article, so they combined a products characterized by really high vitamin D content (salmon – about 15 ug per 100 g) with other, characterized by low or moderate level (e.g. tuna – about 1.7 ug per 100 g – see: https://ndb.nal.usda.gov/ndb/foods/show/15127) – it must be corrected.
Methods:
What do Authors mean by “elaborated” – did each rower individually prepare his meals or was it prepared by qualified workers?
If the distribution is normal, the mean values should be presented (accompanied by SD), but if it is different than normal, the median, accompanied by minimum and maximum values should be presented – it should be specified that distribution is normal if it is.
Results:
The section must be corrected accordingly, taking into account the indicated above problems
Instead of figures Authors should rather present tables as in the present form, it is unable to see the observed values.
If the distribution is normal, the mean values should be presented (accompanied by SD), but if it is different than normal, the median, accompanied by minimum and maximum values should be presented – it should be specified that distribution is normal if it is.
Discussion:
The section must be corrected accordingly, taking into account the indicated above problems
The limitations paragraph should be corrected accordingly.
Conclusions:
The section must be corrected accordingly, taking into account the indicated above problems
Author Response
The authors would like to thank the editor for giving us the opportunity to revise our manuscript, and the reviewers for their thoughtful and constructive comments. The manuscript has been revised thoroughly according to comments and suggestions. Changes to the original manuscript have been with highlight in green. An itemized point-by-point response to the reviewers’ comments is presented below.
Reviewer(s)' Comments to Author:
Reviewer 1
Authors addressed the concerns. Thank you.
The manuscript entitled “Effects of vitamin D supplementation on haematological values and muscle recovery in elite male traditional rowers” presents interesting issue, but some important corrections are necessary.
Major:
Reviewer 1: Authors explained their errors with the vitamin D content in the previous version of the manuscript and corrected them. But it is still not known what was the vitamin D level in the diet at baseline (in the diet prior to the study) – as patients were stated to be vitamin D deficient (Table 1) – it should be explained why.
Answer: Thanks for the comment. The authors do not have the previous data on vitamin D intake. Although not having these data may be a limitation, the data obtained in this study indicate that although the rowers took adequate RDA of vitamin D (15 µg/day), only 50% of those who were supplemented with 3000 IU / day of vitamin D achieved optimal levels of 25 (OH) D for athletic performance enhancement (≥ 50 ng / ml) (table 1). In this sense, Ogan and Pritchett stated that the possibility of athletes requiring increased intake of vitamin D is due to the active use of vitamin D in many metabolic pathways [47]. Authors have included this las sentence in the discussion.
Reviewer 1: Moreover, the intake of food products for 8 weeks of assessment should be presented, as Authors stated that the vitamin D intake was 15-16 ug (Table 2) – it should be presented what was the intake of specific food product groups (fish, meat, milk, dairy beverages, etc.)
Answer: Thanks for your recommendation. The authors have included a table with the ingested rations of different foods in both groups. In this sense, all the information has been added in the results section as in material and methods.
Table 5 shows daily number of portions of different food groups of the elite athletes during study. There was no significant difference between groups (p>0.05), including dietary vitamin D sources such as milk and dairy products (CG: 2.81±0.35 vs. VD3G: 2.76±0.32 servings/day) and fish (CG: 1.89±0.33 vs. VD3G: 1.95±0.41 servings/day).
Table 5. Daily number of portions of different food groups in control group (CG) and vitamin D3 group (VD3G) during 8 weeks of study | |||
CG (n=18) | VD3G (n=18) | p | |
Cereals/grains | 4.26±0.35 | 4.31±0.42 | 0.7004 |
Milk and dairy products | 2.81±0.35 | 2.76±0.32 | 0.6575 |
Fruits | 2.47±0.82 | 2.51±0.72 | 0.8773 |
Vegetables | 2.67±0.65 | 2.47±0.74 | 0.3950 |
Oil and fats | 2.28±0.32 | 2.31±0.41 | 0.8081 |
Legumes | 0.51±0.23 | 0.50±0.21 | 0.8924 |
Dry fruits | 0.42±0.18 | 0.38±0.21 | 0.5436 |
Fish | 1.89±0.33 | 1.95±0.41 | 0.6317 |
Meat and meat products | 1.09±0.45 | 1.00±0.64 | 0.6286 |
Eggs | 0.50±0.01 | 0.50±0.01 | 1.000 |
Sugar and sweets | 0.87±0.25 | 0.77±0.18 | 0.1774 |
Nonalcoholic beverages | 0.27±0.18 | 0.30±0.15 | 0.5905 |
Data are expressed as mean±standard deviation. p: Significantly different between groups by independent t-test | |||
General:
Reviewer 1: The manuscript is in general shabbily prepared (lack of formatting according to recommendations – e.g. references, or abstract, typos – e.g. “descrrbed”)
It seems, that Authors are not native English speakers (e.g. “descrrbed”) – the manuscript should be corrected by native English speaker or preferably by the professional agency.
Answer: Agree with your comment. Manuscript has been corrected by English speaker.
Reviewer 1: The applied abbreviation for vitamin D (VITD) is rather unusual – it should be changed or preferably this abbreviation should be removed
Answer: Thank you for your recommendation. Authors have changed VITD abbreviation for vitamin D through the text.
Reviewer 1: Authors should avoid personal description (e.g. “our research”) and instead use less personal phrases (e.g. “the research”).
Answer: Thank you for this appreciation. Following reviewer consideration authors have changed our study for the study.
Abstract:
Reviewer 1: Authors should precisely indicate the information (e.g. “The supplementation with oral Vitamin D can favor the state of the iron metabolism and testosterone and cortisol as indicator of muscle recovery of the athlete.” – for all athletes or just deficiency ones?)
Answer: Thank you for your commentary. Authors have added “with deficiency” in that sentence. The supplementation with oral Vitamin D can favor the state of the iron metabolism and testosterone and cortisol as indicator of muscle recovery of the athlete with deficiency.
Reviewer 1: “Behaviour” – should be removed
Answer: Thank you for your recommendation. Authors have changed behavior by group-by-time interaction.
Reviewer 1: Lines 45-46 - testosterone and cortisol do not allow to conclude about enhanced muscle recovery – the conclusion should not be associated with the possibility of such relation
Answer: Thank you for your recommendation. Effectively testosterone and cortisol do not allow us to conclude of an improved recovery. However, table 7 shows that serum 25 (OH) D levels are predictors of testosterone cortisol levels. In this sense, the authors have changed the conclusion by: “although it was not sufficient to enhance muscle recovery observed by testosterone and cortisol responses, it observed that serum 25(OH)D levels could be a predictor of anabolic and catabolic hormones”.
Introduction:
Reviewer 1: The section still requires a really major corrections – Authors should properly justify the study, while they should present precisely the specific information with no generalizing and no basic information presented. Moreover, the section should not present all the information that Authors have but only the most important ones, in order to properly justify the study.
Answer: Thanks for your recommendation. The authors have focused on the objective throughout the introduction. They have eliminated little relevant or known information. In addition, the authors have saved information for the discussion, showing only the most important information.
Authors tend to present very basic or even trivial information as a novelty (e.g. “Judd and Tangpricha concluded a significant association among deficient levels of 25-hydroxyvitamin D (…), and the risk of suffering from different types of diseases” – it is well known fact for ages, not concluded only by Judd and Tangpricha but by a number of authors) – such information should be removed as the readers of Nutrients are in general familiar with the role of vitamin D and only more specific information should be presented.
Authors tend to generalize the observations from the referred studies – e.g. they refer the article about cardiovascular diseases, but conclude about “different types of diseases”, etc.
Answer: Thank you for your recommendation. Following this recommendation authors have deleted the next sentence “Judd and Tangpricha concluded a significant association between low levels of 25- hydroxy vitamin D (25(OH)D) (prohormone and metabolite of vitamin D), and the risk of suffering from different types of diseases [1].”
Reviewer 1: In the Introduction section, Authors improperly presented a diet as a source of vitamin D – Authors referred not the databases of nutritional value of products, but some not very prominent article, so they combined a products characterized by really high vitamin D content (salmon – about 15 ug per 100 g) with other, characterized by low or moderate level (e.g. tuna – about 1.7 ug per 100 g – see: https://ndb.nal.usda.gov/ndb/foods/show/15127) – it must be corrected.
Answer: Thanks for your comment. The authors have deleted tuna from the introduction even though some food composition tables have vitamin D values of 7.2 µg per 100 g (see: http://www.bedca.net/bdpub/index_en.php ).
Methods:
Reviewer 1: What do Authors mean by “elaborated” – did each rower individually prepare his meals or was it prepared by qualified workers?
Answer: Thank you for your comment. The authors want to clarify that it was the dietitian-nutritionist of the club who elaborated in an individual way the diets to each rower. In this sense, author have changed the sentence by “Further, the dietitian-nutritionist of the club elaborated a personal diet for each rower.”.
Reviewer 1: If the distribution is normal, the mean values should be presented (accompanied by SD), but if it is different than normal, the median, accompanied by minimum and maximum values should be presented – it should be specified that distribution is normal if it is.
Answer: Thank you for your comment. Authors have added in Statistical Analysis next sentence “As the distribution of all the parameters was normal, the results are presented as means ± standard deviation.”
Results:
Reviewer 1: The section must be corrected accordingly, taking into account the indicated above problems
Answer: Thank you for your commentary. Authors have improved the results section taking into account the above problems.
Reviewer 1: Instead of figures Authors should rather present tables as in the present form, it is unable to see the observed values.
Answer: Thank you for your observation. Authors have changed figures by tables.
Reviewer 1: If the distribution is normal, the mean values should be presented (accompanied by SD), but if it is different than normal, the median, accompanied by minimum and maximum values should be presented – it should be specified that distribution is normal if it is.
Answer: Thank you for your comment. Authors have added in Statistical Analysis next sentence “As the distribution of all the parameters was normal, the results are presented as means ± standard deviation.”
Discussion:
Reviewer 1: The section must be corrected accordingly, taking into account the indicated above problems
Answer: Thank you for your recommendation.
Reviewer 1: The limitations paragraph should be corrected accordingly.
Answer: The authors believe that the section has been corrected according to the problems commented on by the reviewer.
Conclusions:
Reviewer 1: The section must be corrected accordingly, taking into account the indicated above problems
Answer: The authors believe that the section has been corrected according to the problems commented on by the reviewer.

Round 3
Reviewer 1 Report
The manuscript entitled “Effects of vitamin D supplementation on haematological values and muscle recovery in elite male traditional rowers” presents interesting issue, but some important corrections are necessary.
Major:
1. Authors did not indicate what was the vitamin D level in the diet at baseline (at the beginning of the study), while patients were stated to be vitamin D deficient. It should extensively discussed – if Authors did not analyse it, they should at least discuss it as a limitation of the study.
2. The intake of food products for 8 weeks of assessment should be presented, but not in the form that is presented, as the sample size must be known. Authors indicated number of portions, but did not define the portion size. It is crucial to explain the vitamin D intake observed.
General:
The manuscript is in general shabbily prepared (lack of formatting according to recommendations – e.g. references, or abstract)
Authors should avoid personal description (e.g. “we suggest”, “we did not obtain”) and instead use less personal phrases (e.g. “it is suggested”, “it was not obtained”).
Introduction:
In the Introduction section, Authors referred not the databases of nutritional value of products, but not very prominent article – it must be corrected.
Methods:
What do Authors mean by “elaborated” – it seems, that Authors mean rather “calculated”?
Results:
Instead of figures (Figure 1), Authors should rather present tables as in the present form, it is unable to see the observed values.
Discussion:
The section must be corrected accordingly, taking into account the indicated above problems
The limitations paragraph should be corrected accordingly.
Conclusions:
The section must be corrected accordingly, taking into account the recommended vitamin D intake during the study with no information about the intake before the study.